# Gut Microbiota in Different Treatment Response Types of Crohn’s Disease Patients Treated with Biologics over a Long Disease Course

**DOI:** 10.3390/biomedicines13030708

**Published:** 2025-03-13

**Authors:** Xiaolei Zhao, Jun Xu, Dong Wu, Ning Chen, Yulan Liu

**Affiliations:** 1Department of Gastroenterology, Peking University People’s Hospital, Beijing 100044, China; xiaoleibeijing07@163.com; 2Clinical Center of Immune-Mediated Digestive Diseases, Peking University People’s Hospital, Beijing 100044, China; 3Department of Gastroenterology, Peking Union Medical College Hospital, Beijing 100730, China; wudong@pumch.cn

**Keywords:** Crohn’s disease, biological agents, gut microbiota, divergent treatment responses, long disease course

## Abstract

**Background and Aims:** Crohn’s disease (CD) is a chronic inflammatory bowel disease (IBD) with a globally increasing prevalence, partially driven by alterations in gut microbiota. Although biological therapy is the first-line treatment for CD, a significant proportion of patients experience a primary non-response or secondary loss of response over time. This study aimed to explore the differences in gut microbiota among CD patients with divergent long-term responses to biological therapy, focusing on a long disease course. **Methods:** Sixteen CD patients who applied the biological agents for a while were enrolled in this study and were followed for one year, during which fecal specimens were collected monthly. Metagenomic analysis was used to determine the microbiota profiles in fecal samples. The response to biological therapy was evaluated both endoscopically and clinically. Patients were categorized into three groups based on their response: R (long-term remission), mA (mild active), and R2A group (remission to active). The differences in the gut microbiota among the groups were analyzed. **Results:** Significant differences in fecal bacterial composition were observed between the groups. The R2A group exhibited a notable decline in gut microbial diversity compared to the other two groups (*p* < 0.05). Patients in the R group had higher abundances of *Akkermansia muciniphila*, *Bifidobacterium adolescentis*, and *Megasphaera elsdenii*. In contrast, *Veillonella parvula*, *Veillonella atypica*, and *Klebsiella pneumoniae* were higher in the R2A group. **Conclusions:** Gut microbial diversity and specific bacterial significantly differed among groups, reflecting distinct characteristics between responders and non-responders.

## 1. Introduction

Crohn’s disease (CD) is a formidable inflammatory bowel disease (IBD), characterized by a complex interplay of susceptibility genes, environmental influences, and dysregulated mucosal immune responses, gut microbes, and their metabolites in its pathogenesis [1]. The resultant aberrant immune responses to commensal microbes lead to profound damage to the intestinal mucosal layer, marked by extensive epithelial injury, immune infiltration, crypt abscess formation, and chronic inflammation. In recent years, the therapeutic landscape for CD has shifted toward biological agents, including anti-tumor necrosis factor-alpha (TNF-α), anti-integrin therapies, and anti-interleukin agents targeting IL-12/23. Despite the availability of these therapies, significant variability in the treatment response remains evident among patients exhibiting similar clinical or endoscopic manifestations of IBD. Some patients achieve sustained remission, while others experience persistent mild symptoms or fail to respond, remaining in active disease. This variability underscores the pressing need for personalized treatment approaches (refining drug selection), aimed at maximizing beneficial clinical outcomes, improving quality of life, and minimizing healthcare costs.

Previous research has delved into the relationship between disease characteristics and therapeutic responses; however, conclusive findings have been elusive. To date, factors such as age, sex, weight, and smoking status have yet to demonstrate consistent correlations with the efficacy of anti-TNF and other targeted therapies. Notably, recent meta-analyses have suggested that early intervention in CD is associated with enhanced rates of remission and mucosal healing. Interestingly, no significant associations have been found between disease location and responses to anti-TNF or ustekinumab therapies.

Emerging data regarding serum biomarkers have also shown mixed results. A recent retrospective study indicated that a higher ratio of serum triiodothyronine to thyroxine (T3/T4) in elderly patients may be linked to mucosal healing, although no direct correlation to the specific biological agents used was observed. Additionally, C-reactive protein (CRP) levels demonstrated inconsistent patterns to treatment responses; meanwhile, elevated CRP levels in CD patients aligned with improved responses to anti-TNF therapies and yet exhibited a negative correlation in ulcerative colitis (UC) patients. Similar disparities have been noted with vedolizumab but not with ustekinumab.

Moreover, prior investigations have established that the gut microbes of IBD patients often exist in a state of dysbiosis, characterized by reduced microbial diversity, specifically diminished *Firmicutes* and a decreased abundance of *Clostridium perfringens* [2,3,4,5,6], alongside increased rates of *Aspergillus* [3]. Notably, research by Andrews et al. [7] has illustrated that dysbiosis persists in CD patients even during clinical remission. However, the existing studies [8,9,10,11,12,13,14,15] (Table 1) focused on primary non-response and were conducted during the induction phase, with limited attention to the later stages of treatment.

In light of these findings, we designed a pilot study to investigate whether there are distinct gut microbiota among CD patients with different response types during the maintenance phase of biological therapy.

Our study was designed to explore the differences in gut microbiota among CD patients with divergent long-term responses to biological therapy, focusing on a long disease course. Understanding distinct gut microbiota may provide crucial insights into tailored therapeutic strategies and improved management of Crohn’s disease.

## 2. Materials and Methods

### 2.1. Study Population and Ethics

The study cohort consisted of 16 patients diagnosed with CD who received biological therapy at the Department of Gastroenterology, Peking University People’s Hospital, between 2021 and 2022. Participants had been on biological therapy for a duration ranging from 0.5 to 5 years. Stool samples were collected monthly for up to one year for metagenomic analysis. This study was approved by the Ethics Committee of Peking University People’s Hospital (Approval No. 2021PHB255). Written informed consent was obtained from all participants before sample collection, ensuring ethical compliance throughout this study.

### 2.2. Evaluation of Treatment Response

The therapeutic response to biological agents was evaluated through a combination of colonoscopy, laboratory tests, and clinical assessments. The Simple Endoscopic Score for Crohn’s disease (SES-CD) was utilized to assess endoscopic activity. Remission was defined as an SES-CD score of 0–2 points, while mild activity was classified as an SES-CD score of 3–6 points. A score equal to or above 7 points indicated a shift from remission to active disease (R2A). Additional clinical assessments included the modified Harvey–Bradshaw Index to further evaluate disease activity.

### 2.3. Fecal DNA Extraction

During the study period, 192 fecal samples were collected and stored at −80 °C before use [16]. DNA extraction from these samples was performed using a Feces: QIAamp DNA Stool Mini Kit (QIAGEN; Hilden, Germany) according to the manufacturer’s protocols. A Qubit Fluorometer was used to determine the DNA concentration. DNA completeness and purity were checked by running the samples on 1.2% agarose gels (Cat No. 10208, Yeasen; Shanghai, China).

### 2.4. Analysis of Sequencing Data

Metagenomic sequencing data were analyzed to characterize bacterial profile. The sequencing libraries were constructed with a NEBNext^®^ Ultra™ DNA Library Prep Kit for Illumina^®^ (New England Biolabs, Ipswich, MA, USA). The products were purified using Agaros Agencourt AMPure XP (Beckman Coulter Inc., Brea, CA, USA) and quantified using the GenNext™ NGS Library Quantification Kit (Toyobo Co., Ltd., Tokyo, Japan). The libraries were sequenced using the Illumina Novaseq 6000 150 bp paired-end technology at TinyGen Bio-Tech Co., Ltd. (Shanghai, China). The raw fastq files were demultiplexed based on the index. The raw sequences were inspected with FastQC (v0.11.9) [17] and end read bases with quality scores (Qscore) lower than 30 were trimmed and quality-controlled with trimmomatic (v0.39) [18] and KneadData software (v0.6.1) (https://github.com/biobakery/kneaddata, accessed on 12 March 2025). Taxonomic classification was conducted with Kraken2 (v2.0.9-beta) [19]. Kraken2 hits accumulating < 10% K-mers matching the reference sequence were discarded and a hit was considered true only if at least 50 reads were aligned against the reference. Metabolic profiling was performed with HUMAnN2 (v0.9.9) [20]. Output tables were labeled with UniRef90 names using the script human2_rename_table, and gene family abundance was renormalized with the script humann2_renorm_table, from RPK to compositional units (counts per million) to enable between-sample comparisons. Genes were regrouped to functional categories with script humann2_regroup_table, as well as to enzyme commission (EC) categories. Other thresholds that were not clearly demarked were set according to the software’s default parameters.

We first evaluated the bacterial alpha diversity with the Shannon, Simpson, and Chao1 indexes. A principal coordinate analysis (PCoA) diagram based on the Bray–Curtis distance was used for an investigation of bacterial beta diversity. The bacterial community was profiled at different taxonomic levels. The overall distribution level of bacteria was further analyzed. Visualization of the bacterial community was performed through the Pavian Macro Genome data browser’s online server (https://fbreitwieser.shinyapps.io/pavian/, accessed on 12 March 2025). Furthermore, to identify fungal species indicators, all identified fungal genera were analyzed using the indicspecies package [21].

### 2.5. Statistical Analysis

R software (v4.0.1, R Foundation for Statistical Computing; Vienna, Austria) with the ggplot2 (v 3.3.2) package were used for data visualization [22]. Permutational multivariate analysis of variance (PERMANOVA, Adonis test of vegan v 2.5–6) was performed for statistical analysis of beta diversity. The *t*-test and nonparametric Mann–Whitney U test were used to compare two groups (the dataset encompasses twelve longitudinal time points per patient). One-way analysis of variance (ANOVA) and Kruskal–Wallis H nonparametric tests compared the three groups. Spearman’s correlation analysis was performed, and the *p*-value was corrected with the false discovery rate (FDR). A *p*-value or FDR ≤ 0.05 was considered statistically significant.

## 3. Results

### 3.1. Patient Characteristics

A total of 16 patients diagnosed with CD participated in this study. The baseline characteristics of these patients are summarized in Table 2.

### 3.2. Diversity of Gut Microbiota in CD Patients

Participants were classified into three groups based on their clinical status: the Remission (R) group (*n* = 8), the Mild Active (mA) group (*n* = 4), and the Remission to Active (R2A) group (*n* = 4).

The gut microbes represent a complex ecosystem whose diversity reflects its functional capacity. Higher microbial diversity is indicative of a more stable ecological state. Our analysis revealed that the diversity, as measured by both the Shannon and Simpson indexes, was significantly greater in the R and mA groups compared to the R2A group (*p* < 0.05). However, no statistically significant difference in diversity were observed between the R and mA groups. Furthermore, the Chao1 index suggested a trend whereby diversity was higher in the R group compared to the mA and R2A groups (*p* < 0.05), with the mA group also exhibiting higher diversity than the R2A group (*p* < 0.05) (Figure 1).

### 3.3. Differences in Microbiota Composition Among Groups

Bray–Curtis dissimilarity-based PCoA revealed distinct clustering of gut microbiota communities among the three groups. The composition of gut bacterial genera in CD patients is illustrated in Figure 2. Different colors correspond to the respective groups. PCoA analysis identified three community clusters with notable differences. Adonis testing demonstrated statistically significant differences in microbial community composition between the R and mA (*p* < 0.001), mA and R2A (*p* < 0.001), and R and R2A (*p* < 0.001) groups.

### 3.4. Bacterial Genera at the Species Level

To gain a more precise understanding of gut bacterial composition, we examined the specific abundance of bacteria at the species level rather than at the family, order, or genus levels. As shown in Figure 3, the R group (green) displayed significantly higher abundances of *Gordonibacter urolithinfaciens*, *Rhizobium leguminosarum*, *Acidaminococcus fermentans*, and *Clostridium cellulosi*, while *Shigella dysenteriae*, *Shigella flexneri*, *Atlantibacter hermannii*, and *Fusobacterium pseudoperidonticum* were elevated in the R2A group (FDR-corrected p [p FDR] < 0.001). Notably, the abundance of *Belliella baltica*, *Veillonella rodentium*, *and Mucilaginibacter* sp. *HYN0043* were significantly higher in the mA group (FDR-corrected p [pFDR] < 0.0001) compared to *Desulfovibrio vulgaris* and *Paenibacillus mucilaginosus*, which were more abundant in the R group (FDR-corrected p [pFDR] < 0.01). Furthermore, when comparing the mA and R2A groups, *Pontibacter* sp. *SGAir0037* and *Riemerella anatipestifer* showed significantly higher abundance in the mA group (FDR-corrected p [p FDR] < 0.0001) relative to *Atlantibacter hermannii* in the R2A group (FDR-corrected p [pFDR] < 0.05).

### 3.5. Composition of Bacteria at Various Taxonomic Levels

To determine whether bacterial composition differed between the study groups, we categorized the data at the domain, phylum, class, order, family, genus, and species levels (Figure 4). Clear differences were observed among the three groups of CD patients.

In the R group, *Bacteroidetes* and *Firmicutes* were the predominant phyla, with the orders *Bacteroidales* and *Clostridiales* being the most abundant. Specific species, including *Bacteroides vulgatus*, *Faecalibacterium* prausnitzii, *Bacteroides dorei*, and *Bacteroides uniformis*, exhibited higher abundance levels. Conversely, in the mA group, there was a marked increase in *Candidatus Saccharibacteria* and *Proteobacteria* at the phylum level, with a higher abundance of *Prevotella* at the genus level. Additionally, the species richness of *Veillonella parvula* and *Escherichia coli* was elevated in this group.

In the R2A group, there was a decrease in the order Bifidobacteriales, while significant increases in species such as *Bacteroides fragilis*, *Parabacteroides distasonis*, *Lachoclostridium* sp. YL32, *Ruminococcus gnavus*, *Megamonas funiformis*, and *Escherichia coli* were observed compared to the R group.

### 3.6. Focus on Key Bacterial Species in the Three Groups

We further investigated the predominant gut bacteria across the three groups. As depicted in Figure 5, *Akkermansia muciniphila*, *Bifidobacterium adolescentis*, and *Magasphaera elsdenii* were more abundant in the R group. In contrast, the R2A group exhibited higher levels of *Veillonella parvula*, *Veillonella atypica*, and *Klebsiella pneumoniae*. Additionally, *Veillonella parvula* and *Haemophilus parainfluenzae* were found to be increased in the mA group, offering valuable insights into potential key bacterial species that may influence the disease course.

## 4. Discussion

In this study, we investigated the differences in fecal bacterial compositions in a prospective cohort of patients whose CD was treated with biological agents with different treatment responses. While prior research has explored gut microbiota in IBD patients under biological therapy, the majority of these studies focused on the initial treatment responses rather than the long-term effects of sustained therapy. To extend those investigations, this study performed a valuable and comprehensive analysis of distinct gut microbiotas in CD patients with divergent responses to biological treatment over a long time.

### 4.1. Diversity of Gut Microbes

The alterations in gut microbe signal changes in the complex intestinal epithelial micro-environment reflect the ecosystem’s overall functionality. Increased microbial diversity is often associated with a more stable gut environment. Previous studies have consistently shown reduced fecal microbiota diversity in IBD patients compared to healthy individuals, with a noted decline of approximately 25% in taxa diversity indicating dysbiosis correlated with high disease activity [23,24]. Furthermore, investigations have demonstrated that changes in gut microbes and associated metabolites, such as short-chain fatty acids (SCFAs), are influenced by biological therapy [25,26]. Notably, patients with effective responses to anti-TNF-α treatment exhibited higher gut microbiome diversity compared to non-responders [27,28,29,30].

Most previous studies have concentrated on the early phases of biological induction therapy, with limited focus on the maintenance therapy stage. For instance, Seong [31] reported that IBD patients in the mucosal healing group showed greater microbial diversity during maintenance therapy than those in the non-healing group. Similarly, Sanchis-Artero [32] found that responders exhibited higher alpha diversity after six months of biological maintenance treatment. Our analysis highlighted that patients in the remission (R) and mild active (mA) groups demonstrated significantly higher diversity (e.g., Shannon, Simpson, and Chao1 indices) than those in the remission to active (R2A) group. Importantly, the R group exhibited the highest diversity, providing compelling evidence of the ongoing changes in gut microbiome dynamics during maintenance therapy across varying response states. Thus, microbial diversity serves as a crucial marker reflecting therapeutic efficacy.

### 4.2. Specific Microorganisms at the Species Level

#### 4.2.1. *Akkermansia muciniphila*

We observed a significant increase in the abundance of *Akkermansia muciniphila* (AKK) in the R group. Extensive research has linked AKK to various health conditions, including metabolic syndrome, colitis, inflammation, and response to cancer immunotherapies. Specifically, studies indicate that AKK plays a crucial role in enhancing mucus barrier integrity and modulating the activity of regulatory T cells (Tregs) and cytotoxic T lymphocytes. Notably, AKK contributes to the maintenance of gut barrier function, with diminished levels observed in IBD patients compared to healthy individuals [33].

For instance, Wang et al. [34] demonstrated a significant reduction in AKK in mice with ulcerative colitis, further supporting its potential protective role. Additionally, pediatric patients with Crohn’s disease who did not achieve remission exhibited both lower microbiome diversity and decreased abundance of AKK [35]. These findings suggest that AKK may serve as a valuable biomarker for predicting therapeutic efficacy in IBD treatment.

#### 4.2.2. *Faecalibacterium prausnitzii*

Our results demonstrate that the abundance of *Faecalibacterium prausnitzii* was significantly higher in the R group compared to the mA and R2A groups. Numerous studies have established that *Faecalibacterium* serves as an important indicator of gut health, playing a vital role in mitigating toxicity and maintaining immune homeostasis within the gastrointestinal tract.

For instance, Ying et al. [36] found that supplementation with *F. prausnitzii* could alleviate fibrosis in the colons of mice, highlighting its potential therapeutic effects. Furthermore, studies by Lopez-Siles and Henry [37,38] have reported a marked decline in *Faecalibacteriu* populations among patients with IBD. These findings collectively suggest that *Faecalibacterium* might exert a crucial influence on the regulation of regulatory T cells (Tregs), which are known to significantly impact the pathogenesis of IBD.

#### 4.2.3. *Bifidobacterium adolescentis*

*Bifidobacterium adolescentis* exhibited the highest abundance in the R group. Yang et al. [39] observed that *B. adolescentis* levels were significantly elevated in healthy mice compared to those with DSS-induced colitis. This specific bacterium has been shown to ameliorate colitis through the modulation of Treg and Th2 responses. Additionally, Lina Fan [40] demonstrated that *B. adolescentis* can alleviate colitis in mice by inducing a Treg/Th2 response. Furthermore, Kinga Kowlska [41] reported a reduction in *B. adolescentis* in newly diagnosed Crohn’s disease (CD) patients when compared to healthy children.

In our study, we also noted that patients in remission exhibited the highest abundance of *B. adolescentis*, suggesting a positive association between this bacterium and the efficacy of biological therapies.

#### 4.2.4. *Prevotella*

There is limited evidence regarding the role of *Prevotella* in IBD. Zhang et al. [42] reported that *Prevotella* spp. were more prevalent in patients with active CD compared to those in remission. This finding aligns with our observations, which indicated a significant increase in *Prevotella* abundance in the mA group compared to the R group. Conversely, Xiaoxiao Fan [43] demonstrated that *Prevotella histicola* can ameliorate DSS-induced colitis via the NF-kB signaling pathway. Therefore, the relationship between *Prevotella* and IBD warrants further investigation.

#### 4.2.5. *Veillonella parvula*

In our study, *Veillonella parvula* was found to be enriched in both the mA and R groups. This oral microbe is associated with inflammation through nitrate- and lactate-dependent metabolic pathways and has been reported to be elevated in the intestines of patients with IBD [44]. Investigating how *Veillonella parvula* colonizes the intestine presents a valuable opportunity to explore the oral–gut axis in the context of IBD.

#### 4.2.6. *Klebsiella pneumoniae*

*Klebsiella pneumoniae* is a member of the extended-spectrum β-lactamase-producing *Enterobacterales*. Sara Federici [45] identified a clade of *K. pneumoniae* that was strongly associated with disease severity. Additionally, Ayda AK [46] found that *K. pneumoniae* colonized more than half of the IBD patients studied, correlating with increased disease activity. Our findings align with these observations, as we noted a higher prevalence of *K. pneumoniae* in the R2A group. This suggests that *K. pneumoniae* may act as a potential trigger for antibiotic-induced disturbances in gut microbes.

### 4.3. Limitations of the Research

This study has several limitations that should be acknowledged.

First, the sample size is relatively small. This was primarily due to the exploratory nature of the study, which was designed as a pilot study to preliminarily investigate the relationship between gut microbiota and the treatment response in IBD patients. The small cohort size was further influenced by the rarity of CD, the challenges of longitudinal follow-up over one year, and the impact of the COVID-19 pandemic on patient recruitment and retention. As a result, the statistical power of the analyses may be limited, and the generalizability of the findings should be interpreted with caution.

Second, while we controlled for some confounding factors, others such as antibiotics, diet, lifestyle, and environmental influences were not rigorously monitored. These factors may have contributed to variability in the gut microbiota composition, potentially affecting the observed results.

Third, the absence of a healthy control group limits our ability to distinguish disease-specific microbial changes from other influences. Without a control group, it is challenging to determine whether the observed microbial shifts are unique to IBD patients or reflect broader variations in gut microbiota.

Finally, this study is primarily descriptive and does not provide mechanistic insights into the observed microbial changes, such as exploring the metabolic and immune pathways associated with key bacterial taxa (e.g., *Akkermansia muciniphila*). Studies taking that approach are needed to establish causal relationships and deepen our understanding of the gut microbiota’s role in treatment response.

### 4.4. Future Directions

Based on this pilot study, we suggest a scientifically rigorous clinical research protocol to investigate the association between gut microbiota and biological therapy in IBD patients, as outlined in Figure 6.

### 4.5. Clinical Implications of Microbial Differences

The findings of this study have important clinical implications. By identifying microbial signatures associated with different treatment responses, this research paves the way for predictive biomarkers to guide personalized therapy in IBD. For example, gut microbiota profiling could be used to stratify patients into subgroups based on their likelihood of responding to specific biologic therapies, enabling earlier and more targeted interventions. Furthermore, interventions such as fecal microbiota transplantation (FMT) or targeted modulation of key bacterial taxa could be explored as adjunct therapies to improve treatment outcomes and patient prognoses. These advances would represent a significant step toward precision medicine in IBD management.

## 5. Conclusions

Most studies concentrate on the microbiome during the induction phase of therapy, seeking to identify precise biomarkers. Our study offers novel insights into the maintenance phase, investigating the relationship between gut microbes and disease response. We found that microbial diversity and specific bacterial populations significantly differed among response groups, reflecting distinct characteristics between responders and non-responders. Notably, we identified differences in the abundance of several bacterial species, including *Akkermansia muciniphila*, *Faecalibacterium prausnitzii*, *Bifidobacterium adolescentis*, *Prevotella*, *Veillonella parvula*, and *Klebsiella pneumoniae*, which may serve as potential biomarkers for predicting the efficacy of biologic agents in IBD. This research represents a promising step toward precision medicine.

## Figures and Tables

**Figure 1 biomedicines-13-00708-f001:**
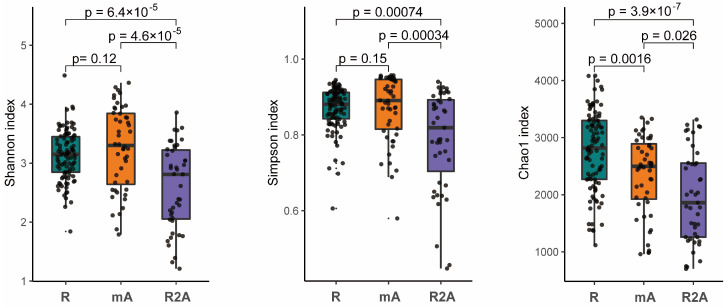
The Shannon, Simpson, and Chao1 indexes, representing species diversity, indicate that diversity was significantly higher in the R and mA groups compared to R2A (*p* < 0.05). No statistically significant difference was observed between R and mA in the Shannon and Simpson indices. However, the Chao1 index suggested a trend of highest diversity in the R group and the lowest in R2A, with a statistically significant difference noted among the groups.

**Figure 2 biomedicines-13-00708-f002:**
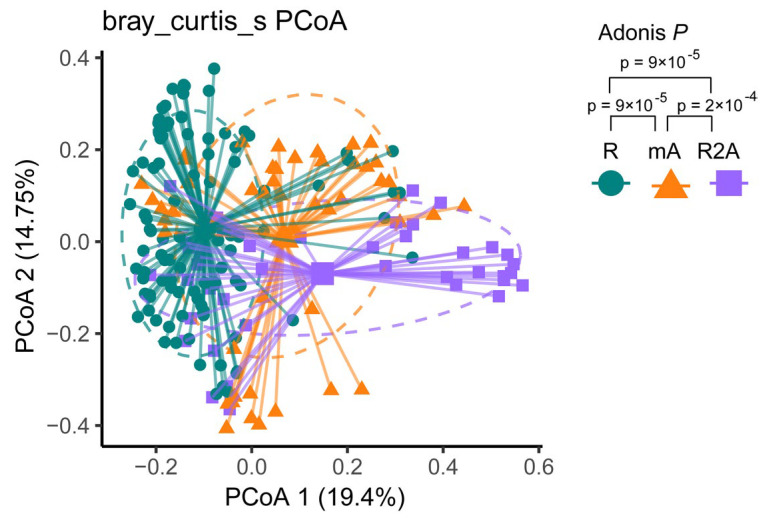
The green spot represents the R group, the orange triangle represents the mA group, and the purple square represents the R2A group. Principal coordinates analysis (PCoA) illustrated distinct subtypes among the groups. Adonis analysis revealed a statistically significant dissimilarity between the groups (*p* < 0.001).

**Figure 3 biomedicines-13-00708-f003:**
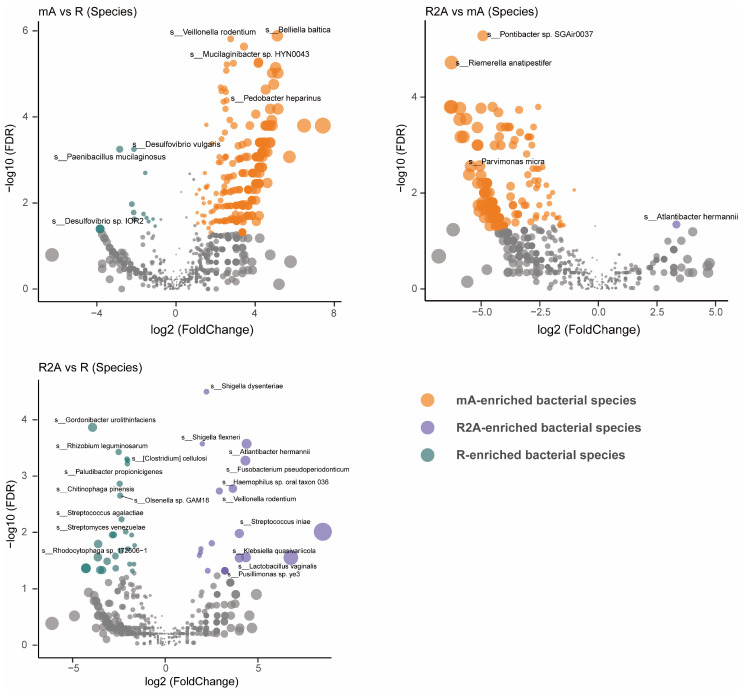
Comparison of gut bacterial composition and differences at the species level among three groups of CD patients.

**Figure 4 biomedicines-13-00708-f004:**
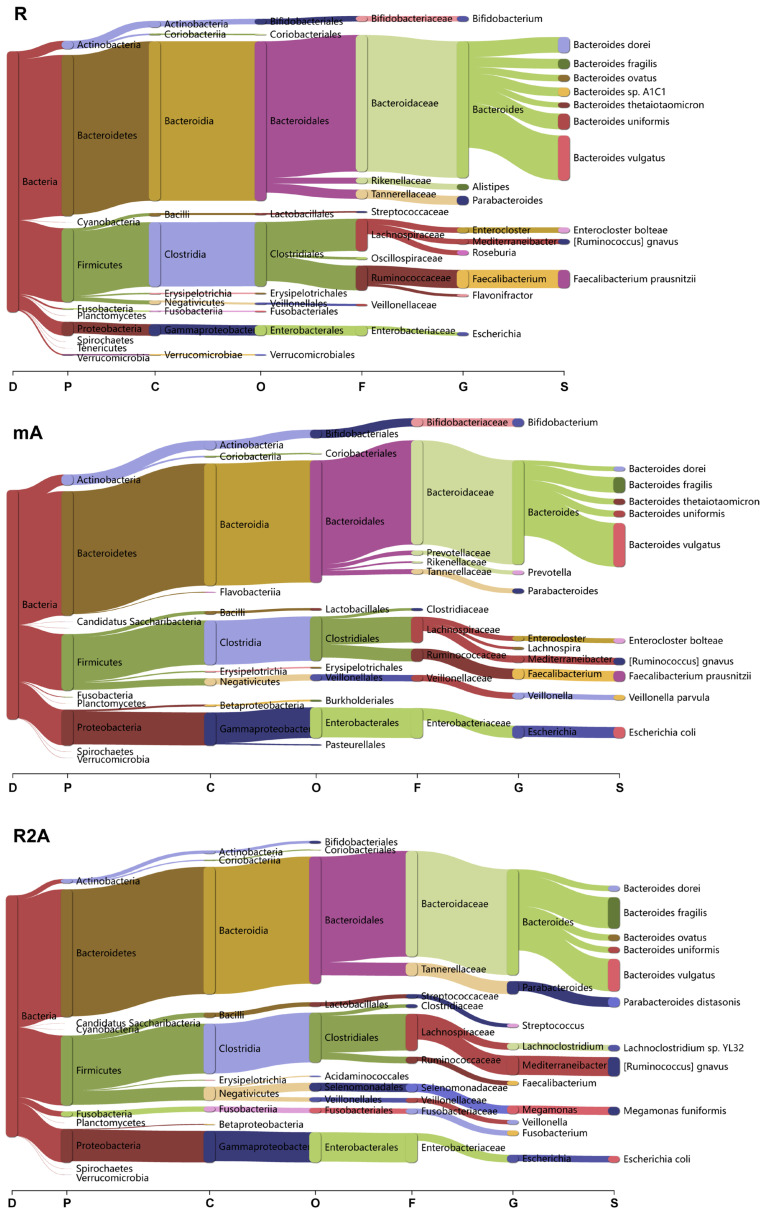
Analysis of bacterial composition and changes across various taxonomic levels (Domain, Phylum, Class, Order, Family, Genus, Species) among the R, mA, and R2A groups.

**Figure 5 biomedicines-13-00708-f005:**
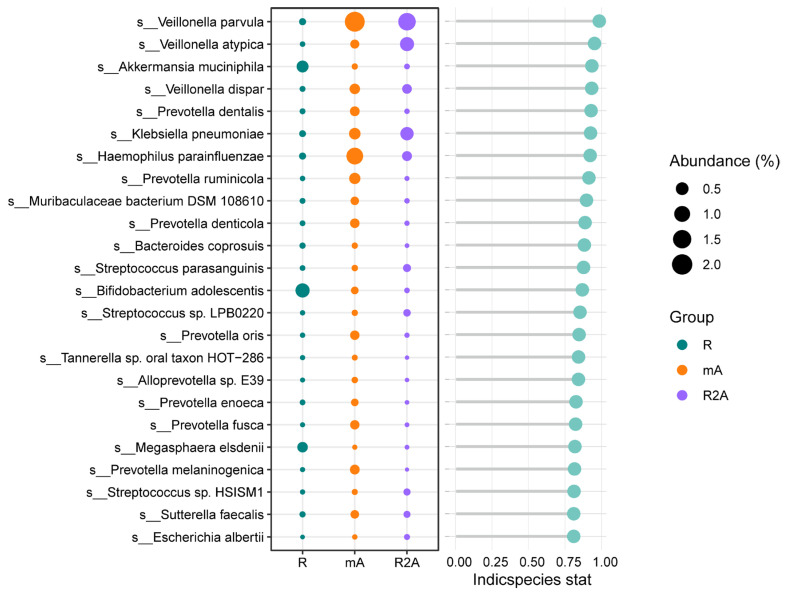
Differences in the abundance of specific bacterial species among the three groups.

**Figure 6 biomedicines-13-00708-f006:**
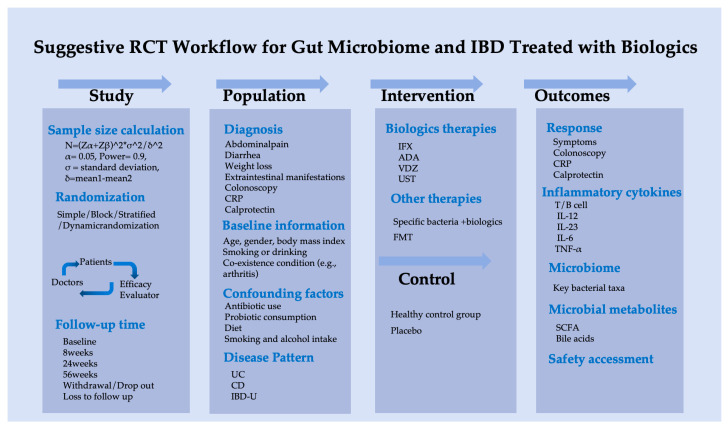
Suggestive randomized controlled trial workflow of gut microbiome for IBD treated with biologics.

**Table 1 biomedicines-13-00708-t001:** Summary of research focused on gut microbes and IBD biologic therapy.

Study	Study Type	Year	Country	No. of IBD Patients		Medical Therapy	Sample	Fecal Collection Intervals	Time of Follow-Up	Sequencing Method	Result
				Total	Subgroups						
Rebecka et al. [8]	Prospective	2021	Finland	72	25CD 47UC	IFX	fecal	0 (baseline), 2, 6, and 12 weeks	1 year	16s rRNA	differed before the start of the IFX non-response: *Clostridia* ↓ *Candida* ↑
Ananthakrishnan et al. [9]	Prospective	2017	USA	85	43UC 42CD	VDZ	fecal	0 (baseline), 6, 14, 30, and 54 weeks	1 year	Meta- genomics	responsive CD patients: *Roseburia inulinivorans* ↑ *Burkholederiales* ↑ and branched-chain amino acid synthesis enriched
Aden et al. [10]	Prospective	2019	Germany	23	13UC 10CD	IFX/ADA	fecal	0 (baseline), 2, 6, and 30 weeks	30 weeks	16s rRNA	diversity indices did not vary in remission, but in remission, butyrate and substrate were associated
Ding et al. [11]	Prospective	2020	UK	76	76CD	IFX/ADA	fecal	0 (baseline), 3 monthly	1 year	16s rRNA	responsive CD: higher deoxycholic acid; non-responsive CD: sulfate and glycine-conjugated primary bile acids
Effenberger et al. [12]	Prospective	2021	Australia	65	43CD 22UC	IFX/ADA	fecal	0 (baseline), 12, 30 weeks	30 weeks	16s rRNA	remission: *Proteobacteria* ↓ *Bacteroidetes* ↑ higher butyrate
Colman et al. [13]	Prospective	2023	USA	74	52 CD 21 CD 1 IBD-U	VDZ	fecal	0 (baseline), 2, 14 weeks, 6 month, 1year	1 year	Meta- genomics	early response: *Firmicute A. hadrus* ↑ abundance of butyrate
Jiang et al. [14]	Prospective	2024	China	21	21 UC	VDZ	fecal	0 (baseline), 14 weeks	14 weeks	16s rRNA	after treatment: *bifidobacterium* ↑ *bacteroides sartorii* ↑ early remission: combined acetamide, taurine, and putrescine
Busquets et al. [15]	Prospective	2021	Spain	38	14CD 24UC	IFX/ADA	fecal	0 (baseline) and monthly	1 year	16s rRNA	greater discriminatory: *M. smithi* (*MSM*), *A. muciniphila* (*AKK*), and *F. prausnitzii phylogroup2* (*PHGII*)

**Table 2 biomedicines-13-00708-t002:** The baseline clinical characteristics of CD patients who participated.

No.	Gender	Montreal Classification	Group	Disease Course (y)	Biologic Agent	Concomitant Medication
1	F	A2 L3 + L4 B1	R2A	5	IFX	AZA
2	M	A2 L2 + L4 B1	R	3	IFX	steroid
3	F	A2 L3 B3	R	5	IFX	-
4	M	A2 L3 + L4 B1	mA	2	IFX	AZA
5	M	A2 L2 B1	R	3	IFX	-
6	M	A2 L2 B2	mA	1	IFX/ADA	-
7	M	A2 L1 B2	R2A	5	IFX/UST	-
8	M	A2 L2 B3	mA	2	ADA	-
9	M	A1 L3 + L4 B3	mA	1	ADA	-
10	M	A1 L3 B1	R	2	IFX	-
11	M	A2 L2 B3	R	2	IFX	5-ASA
12	F	A2 L3 B1	R	1	IFX	-
13	M	A2 L3 B3	R	2	IFX	AZA
14	M	A2 L3 B1	R	0.5	IFX	-
15	M	A2 L3 B3	R2A	0.5	IFX/ADA	-
16	M	A1 L3 B3	R2A	0.5	IFX	-

(F: female; M: male; IFX: infliximab; ADA: adalimumab; UST: ustekinumab; AZA: azathioprine; 5-ASA: 5-aminosalicylic acid).

## Data Availability

The raw data supporting the conclusions of this article will be made available by the authors on request.

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
