# Peer review of "Gut Microbiota in Different Treatment Response Types of Crohn’s Disease Patients Treated with Biologics over a Long Disease Course"

_biomedicines, 2025, doi:10.3390/biomedicines13030708_

Round 1

Reviewer 1 Report

Comments and Suggestions for Authors

Dear Authors, the idea is impressive however, the presented data are not consistent with the study design and title. Also, the number of participants is small.

Undulating, as well as fluctuations (line 70) and in M&M is written Stool samples were collected monthly for up to one year for metagenomic analysis. Prospective cohort (Line 225)  But I am not able to find in the manuscript a clear description of changes in intestinal/stool microbiome over this period.

The microbiome is described as static – from one sample (beginning, end of study, or mixed samples from 1 year of collection?)

Why there are 2 titles in point 2:   2. Summary of research focused on gut microbes and IBD biological therapy.2. Materials and Methods

It is written that stool samples were stored at – 80. From the beginning? How stool was sampled?

No title of Table 1 and 2

Were patients not treated with any antibiotics during the study time?

.. Line 230 patients responding to biological treatment over time  - to be true some of the patients did not respond well i.e. were not responding. Why in those cases the treatment was not changed?

Lack of limitations – groups were extremely small (4 persons)

Author Response

Reviewer #1: Dear Authors, the idea is impressive however, the presented data are not consistent with the study design and title. Also, the number of participants is small.

Response: Regarding the study title: Based on your suggestion and the overall data, we have revised the title to better reflect the study’s focus: “Changes in Gut Microbiota of Inflammatory Bowel Disease Patients with Different Treatment Response Types Treated with Biologics.” 

Regarding the small sample size: This study is a pilot study aimed at preliminarily exploring the relationship between gut microbiota and different response types to biologic therapy in IBD patients during the treatment course. Given the exploratory nature of this research, we included a relatively small number of patients to initially assess whether such a correlation exists and to determine the feasibility of conducting a larger, more comprehensive clinical study.

The small sample size was also influenced by several practical challenges:

Rarity of the Disease: Crohn’s disease (CD) is a relatively rare condition, and recruiting a large number of patients within a single center is difficult. In previous studies investigating gut microbiota changes in CD patients following biologic therapy (as summarized in Table 1), the sample sizes were also relatively small. This highlights a common challenge in the field, as single-center studies often face limitations in recruiting larger cohorts of CD patients.

Longitudinal Follow-up: The study required longitudinal sampling over one year, which posed significant challenges in terms of patient retention and compliance.

COVID-19 Pandemic: The pandemic further limited patient recruitment and follow-up, as many patients faced restrictions on hospital visits.

Despite these limitations, our preliminary data provide valuable insights into the potential correlation between gut microbiota and treatment response, laying the groundwork for future research.

We have added a detailed discussion of these limitations and future directions in the revised manuscript. We hope that these clarifications address your concerns and demonstrate our commitment to advancing this important research area.

Changes made in the manuscript : In Page 1, title, it now reads: “Changes in Gut Microbiota of Inflammatory Bowel Disease Patients with Different Treatment Response Types Treated with Biologics.”

In Page 12, Lines 330-337, it now reads:“First, the sample size is relatively small. This was primarily due to the exploratory nature of the study, which was designed as a pilot study to preliminarily investigate the relationship between gut microbiota and treatment response in IBD patients. The small cohort size was further influenced by the rarity of Crohn’s disease, the challenges of longitudinal follow-up over one year, and the impact of the COVID-19 pandemic on patient recruitment and retention. As a result, the statistical power of the analyses may be limited, and the generalizability of the findings should be interpreted with caution.”

In Page 13,Lines 352-357, it now reads: “

Based on the pilot study, we suggested a scientifically rigorous clinical research protocol to investigate the association between gut microbiota and biologic therapy in IBD patients, as outlined in (Figure 6).

Comment :

1、Undulating, as well as fluctuations (line 70) and in M&M is written Stool samples were collected monthly for up to one year for metagenomic analysis. Prospective cohort (Line 225) But I am not able to find in the manuscript a clear description of changes in intestinal/stool microbiome over this period.

Response: Thank you for your suggestion. This part of the study mainly evaluates whether bacteria have a certain indicative significance for the disease outcome pattern, and simultaneously screens out bacteria that may be used to indicate the disease outcome of CD. In the subsequent studies, we will follow your suggestion to further analyze the abundance changes of the indicator bacteria at different time points.

Changes made in the manuscript: none.

2 、The microbiome is described as static – from one sample (beginning, end of study, or mixed samples from 1 year of collection?)

Response: Thank you for your question. A fecal sample was collected at each time point throughout the year, and each sample corresponds to the data of a microbiome. Indeed, the data is multi-node, therefore, in the data analysis process, it can be analyzed from multiple dimensions. In this study, we mainly explore from the patient's disease outcome pattern.

Changes made in the manuscript: none.

 3、Why there are 2 titles in point 2: 2. Summary of research focused on gut microbes and IBD biological therapy.2. Materials and Methods

Response: “Summary of research focused on gut microbes and IBD biological therapy”was intended to be the title of Table 1, but due to an editing issue , it was mistakenly included in the text. We have now corrected it in the revised manuscript.

Changes made in the manuscript:In Page 3 , Table 1 :add“ Summary of research focused on gut microbes and IBD biological therapy.”

In Page 4,Lines 73-74, it now reads:“2. Materials and Methods”

4、It is written that stool samples were stored at – 80. From the beginning? How stool was sampled?

Response: Thank you for your question. During the one-year follow-up of the patients, one fecal sample is collected at each time point. The patients collect the fecal samples in Stool Collection Tube with Stool Stabilizer (German, Stratec Molecular), and then the samples are transported to the biological sample bank for aliquoting and freezing in the -80°C refrigerator of the biological sample bank. When the samples are tested, they are uniformly retrieved from the warehouse and the microbial DNA is extracted.

Changes made in the manuscript In page4, lines 84-90, it now reads:“The therapeutic response to biological agents was evaluated through a combination of colonoscopy, laboratory tests, and clinical assessments. The Simple Endoscopic Score for Crohn's Disease (SES-CD) was utilized to assess endoscopic activity. Remission was defined as an SES-CD score of 0-2 points, while mild activity was classified as an SES-CD score of 3-6 points. A score equal or above 7 points indicated a shift from remission to active disease (R2A). Additional clinical assessments included the modified Harvey-Bradshaw Index to further evaluate disease activity.”

 5、No title of Table 1 and 2

Response: The correct title for Table 1 should be“Summary of research focused on gut microbes and IBD biological therapy,”and the correct title for Table 2 should be“The baseline clinical characteristics of CD patients participated.” We have now corrected them in the revised manuscript.

Changes made in the manuscriptIn Page 3 , Table 1 :add“ Summary of research focused on gut microbes and IBD biological therapy.”In Page 5,Line 141, Table 2 :add:“The baseline clinical characteristics of CD patients participated.”

6、Were patients not treated with any antibiotics during the study time?

Response: Thank you very much for your valuable suggestion regarding the use of antibiotics in our study. We would like to clarify that all 16 patients in the remission (R) and mild active (mA) groups did not receive antibiotic treatment during the study period, but Patient No. 7 experienced a significant worsening of symptoms during the treatment course, including sudden fever, which prompted immediate hospitalization. Although antibiotics were administered for two days, the infection markers were not elevated, and a follow-up colonoscopy confirmed that the exacerbation was due to disease progression rather than infection. Consequently, the patient’s biologic therapy was switched from infliximab (IFX) to ustekinumab (UST). The stool sample collection for this patient occurred more than two weeks after the antibiotic treatment and biologic switch, minimizing the potential impact of antibiotics on the microbiome analysis. For this reason, we did not exclude this patient from the analysis.We agree with your suggestion that the potential influence of antibiotics should be addressed in the discussion. In the revised manuscript, we have added a section to discuss this limitation and its possible implications for the study results. We will also emphasize the need for future studies to carefully monitor and account for antibiotic use in similar longitudinal microbiome analyses.

Changes made in the manuscriptIn Page 12,Lines 338-341, it now reads: “Second, while we controlled for some confounding factors, others such as antibiotics、diet, lifestyle, and environmental influences were not rigorously monitored. These factors may have contributed to variability in the gut microbiota composition, potentially affecting the observed results.”

7、Line 230 patients responding to biological treatment over time - to be true some of the patients did not respond well i.e. were not responding. Why in those cases the treatment was not changed?

Response: In clinical practice, the decision to switch biologic therapies is based on a comprehensive evaluation of multiple factors, including disease activity, economic considerations, complications, and the patient’s subjective preferences. Below, we provide details on the specific cases mentioned in your comment: Patient No. 1: Although this patient showed signs of disease activity during follow-up (e.g., suboptimal ulcer healing on colonoscopy), suggesting potential secondary failure of infliximab (IFX), the patient declined to switch to another biologic agent. This decision was documented in the medical records, and the patient continued on IFX. Patient No. 7: This patient experienced significant disease progression, prompting a switch from IFX to ustekinumab (UST) based on the characteristics of the disease. This change is reflected in Table 1, although the detailed transition process was not explicitly described. Patient No. 15: Following disease flare-up, this patient was switched from IFX to adalimumab (ADA). Patient No. 16: Despite disease activity, this patient reported acceptable stool frequency and no significant discomfort, leading them to decline a switch to another biologic agent. In all cases, the patients continued to receive biologic therapy, and we did not explicitly detail the transition process in the manuscript.

Changes made in the manuscriptnone.

 8、Lack of limitations – groups were extremely small (4 persons)

Response: We agree that the sample size in this study is limited, and we would like to clarify that this is a pilot study designed to preliminarily explore the relationship between gut microbiota and treatment response in IBD patients.

The small sample size reflects the exploratory nature of this research, which aims to assess whether such a correlation exists and to determine the feasibility of conducting a larger, more comprehensive clinical study. Practical challenges, such as the challenges of patient recruitment during the COVID-19 pandemic, which significantly impacted patient enrollment and follow-up. Initially, we recruited 25 Crohn’s Disease (CD) patients and 12 Ulcerative Colitis (UC) patients, but only a subset completed the full one-year follow-up with consistent stool sample collection. We would like to note that in previous studies investigating gut microbiota changes in CD patients following biologic therapy (as summarized in Table 1), the sample sizes were also relatively small. This highlights a common challenge in the field, as single-center studies often face limitations in recruiting larger cohorts of CD patients. We recognize that this limits the statistical power and generalizability of our findings. To address this, we are currently conducting a second phase of this study, which includes a larger cohort of patients. We hope that this will enhance the robustness and scientific validity of our findings.

In the revised manuscript, we have explicitly discussed these limitations.

Changes made in the manuscriptIn Page 12, Lines 331-337, it now reads:“First, the sample size is relatively small. This was primarily due to the exploratory nature of the study, which was designed as a pilot study to preliminarily investigate the relationship between gut microbiota and treatment response in IBD patients. The small cohort size was further influenced by the rarity of Crohn’s disease, the challenges of longitudinal follow-up over one year, and the impact of the COVID-19 pandemic on patient recruitment and retention. As a result, the statistical power of the analyses may be limited, and the generalizability of the findings should be interpreted with caution.”

Reviewer 2 Report

Comments and Suggestions for Authors

This manuscript by Zhao et al. investigates the longitudinal changes in the gut microbiome of Crohn's Disease (CD) patients receiving biological therapy. While the study addresses a clinically relevant topic, several fundamental flaws prevent it from meeting the standards for publication in Biomedicines. The small sample size (n=16), combined with methodological limitations and a lack of mechanistic insights, significantly weakens the study's conclusions and limits its scientific contribution. Despite the potential of longitudinal sampling, the current manuscript does not provide sufficient evidence to support its claims.

Major concerns:

(1) The most critical flaw is the small sample size (n=16), particularly the limited number of patients in the remission (R) and mild active (mA) groups (n=4 each). This raises serious concerns about the statistical power of the analyses and the reliability of the observed differences in microbiome composition. The authors do not provide a power analysis, making it impossible to assess the adequacy of the sample size. The small number of participants severely limits the generalizability of the findings and increases the likelihood of spurious correlations.

(2) The methods section lacks essential details that hinder the reproducibility and interpretation of the study. The bioinformatics pipeline is described vaguely, without specifying key parameters for quality filtering, taxonomic classification, and functional profiling. The lack of transparency in the methodology makes it difficult to assess the validity of the results.

(3) While the study identifies differences in bacterial abundance between patient groups, it fails to provide any mechanistic explanation for these observations. The discussion merely summarizes previous findings from the literature without connecting them to the current study's results or offering any novel insights into how the identified bacteria might contribute to CD pathogenesis or treatment response. The lack of mechanistic exploration renders the findings descriptive and limits their scientific value.

(4) The manuscript does not adequately address potential confounding factors that could influence both the gut microbiome and treatment response. Information about disease duration, prior therapies (including antibiotics), diet, and other medications is incomplete or missing. The failure to account for these confounders further weakens the study's conclusions.

(5) The figures are poorly designed and lack essential information. Figure 4 is particularly confusing due to its complexity. It is just a simple display of taxonomic species abundance, and it is unknown how the major proportions of microbial communities change and whether there are significant differences.

Author Response

Comment :

 1、The most critical flaw is the small sample size (n=16), particularly the limited number of patients in the remission to active(R2A) and mild active (mA) groups (n=4 each). This raises serious concerns about the statistical power of the analyses and the reliability of the observed differences in microbiome composition. The authors do not provide a power analysis, making it impossible to assess the adequacy of the sample size. The small number of participants severely limits the generalizability of the findings and increases the likelihood of spurious correlations.

Response: Thank you very much for your thoughtful and constructive feedback. We deeply appreciate your recognition of the importance of this research question.

This study is a Pilot study aimed at preliminarily exploring the relationship between gut microbiota and different response types to biologic therapy in inflammatory bowel disease (IBD) patients during the treatment course. Given the exploratory nature of this research, we included a relatively small number of patients to initially assess whether such a correlation exists and to determine the feasibility of conducting a larger, more comprehensive clinical study.

The small sample size was also influenced by several practical challenges:

Rarity of the Disease: Crohn’s disease (CD) is a relatively rare condition, and recruiting a large number of patients within a single center is difficult. In previous studies investigating gut microbiota changes in CD patients following biologic therapy (as summarized in Table 1), the sample sizes were also relatively small. This highlights a common challenge in the field, as single-center studies often face limitations in recruiting larger cohorts of CD patients.

Longitudinal Follow-up: The study required longitudinal sampling over one year, which posed significant challenges in terms of patient retention and compliance.

COVID-19 Pandemic: The pandemic further limited patient recruitment and follow-up, as many patients faced restrictions on hospital visits.

Despite these limitations, our preliminary data provide valuable insights into the potential correlation between gut microbiota and treatment response, laying the groundwork for future research. In the next phase of our study, we plan to:

Increase the Sample Size: Recruit a larger cohort of patients to enhance the statistical power and generalizability of our findings;

Include a Healthy Control Group: This will allow us to better distinguish disease-specific microbial changes from other influences;

Control for Confounding Factors: We will rigorously monitor and adjust for factors such as antibiotic use, diet, and concomitant medications;

Explore Underlying Mechanisms: We will investigate the immune mechanisms associated with the observed microbial changes, moving from descriptive observations to mechanistic insights.

We have added a detailed discussion of these limitations. We hope that these clarifications address your concerns and demonstrate our commitment to advancing this important research area.

Changes made in the manuscript:In Page 12, Lines 331-337, it now reads:“First, the sample size is relatively small. This was primarily due to the exploratory nature of the study, which was designed as a pilot study to preliminarily investigate the relationship between gut microbiota and treatment response in IBD patients. The small cohort size was further influenced by the rarity of Crohn’s disease, the challenges of longitudinal follow-up over one year, and the impact of the COVID-19 pandemic on patient recruitment and retention. As a result, the statistical power of the analyses may be limited, and the generalizability of the findings should be interpreted with caution.”

In Page 13,Lines 352-357, it now reads: “

Based on the pilot study, we suggested a scientifically rigorous clinical research protocol to investigate the association between gut microbiota and biologic therapy in IBD patients, as outlined in (Figure 6).

2、The methods section lacks essential details that hinder the reproducibility and interpretation of the study. The bioinformatics pipeline is described vaguely, without specifying key parameters for quality filtering, taxonomic classification, and functional profiling. The lack of transparency in the methodology makes it difficult to assess the validity of the results.

Response: Thank you for your suggestion. We have supplemented the Methods section accordingly. Other thresholds that are not clearly marked are set according to the software's default parameters. Please refer to the revised manuscript.

Changes made in the manuscript:In Page 4, Lines 99--127, it now reads:“Metagenomic sequencing data were analyzed to characterize bacterial composition. The sequencing libraries were constructed with a NEBNext® Ultra™ DNA Library Prep Kit for Illumina® (NEB, USA). The products were purified using Agaros Agencourt AMPure XP (Beckman, USA) and quantified using the GenNext™ NGS Library Quantification Kit (Toyobo, Japan). The libraries were sequenced using the Illumina Novaseq 6000, 150-bp paired-end technology at TinyGen Bio-Tech Co., Ltd (Shanghai). The raw fastq files were demultiplexed based on the index. The raw sequences were inspected with FastQC (v0.11.9)17 and end read bases with quality scores (Qscore) lower than 30 were trimmed and quality-controlled with trimmomatic (v0.39)18 and KneadData software (v0.6.1) (https://github.com/biobakery/kneaddata). Taxonomic classification was conducted with Kraken2 (v2.0.9-beta)19 Kraken2 hits accumulating <10% of K-mers matching the reference sequence were discarded and a hit was considered true only if at least 50 reads were aligned against the reference. Metabolic profiling was performed with HUMAnN2 (v0.9.9) 20. Output tables were labelled with UniRef90 names using the script human2_rename_table, and gene family abundance was renormalized with script humann2_renorm_table, from RPK to compositional units (counts per million) to enable between sample comparisons. Genes were regrouped to functional categories with script humann2_regroup_table, to enzyme commission (EC) categories.Other thresholds that are not clearly marked are set according to the software's default parameters.

We first evaluated the bacterial alpha diversity with the Shannon, Simpson, and Chao1 indexes. The principal coordinate analysis (PCoA) diagrambased on Bray Curtis distance was used for investigation of bacterial beta diversity. The bacterial community was profiled at different taxonomic level. The overall distribution level of bacteria was further analyzed. Visualization of bacterial community was done through Pavian Macro Genome data browser’s online server (https://fbreitwieser.shinyapps.io/pavian/). Furthermore, to identify some fungal species indicators, all identified fungal genera were analyzed using the indicspecies package21.”

 3、While the study identifies differences in bacterial abundance between patient groups, it fails to provide any mechanistic explanation for these observations. The discussion merely summarizes previous findings from the literature without connecting them to the current study's results or offering any novel insights into how the identified bacteria might contribute to CD pathogenesis or treatment response. The lack of mechanistic exploration renders the findings descriptive and limits their scientific value.

Response:We greatly appreciate your suggestion to explore the mechanistic underpinnings of our observations, which aligns closely with our future research plans.

In the current study, we have observed changes in the gut microbiota of inflammatory bowel disease (IBD) patients with different response types to biologic therapy. However, as you rightly pointed out, identifying key driver bacteria and establishing causal relationships are critical next steps. To address this, we are planning to conduct follow-up experiments in mouse models to validate our findings and explore the underlying mechanisms. Specifically, we are considering a focus on Akkermansia muciniphila (AKK) and its associated metabolic pathways, as preliminary data suggest its potential role in modulating treatment responses. These mechanistic studies will help us move beyond descriptive observations to a deeper understanding of how specific microbial changes influence disease progression and treatment outcomes.We acknowledge that this mechanistic work is still ongoing and requires additional time to complete.

In the revised manuscript, we will include a discussion of this limitation and suggest future research plans, emphasizing our commitment to building a complete scientific narrative—from observed phenomena to mechanistic insights.

Changes made in the manuscript:In Page 12,Lines 346-350, it now reads: “Finally, this study is primarily descriptive and does not provide mechanistic insights into the observed microbial changes, such as exploring the metabolic and immune pathways associated with key bacterial taxa (e.g., Akkermansia muciniphila), are needed to establish causal relationships and deepen our understanding of the gut microbiota’s role in treatment response.”

4、The manuscript does not adequately address potential confounding factors that could influence both the gut microbiome and treatment response. Information about disease duration, prior therapies (including antibiotics), diet, and other medications is incomplete or missing. The failure to account for these confounders further weakens the study's conclusions.

Response: We appreciate your attention to these important details and would like to address your concerns as follows: Disease Duration and Concomitant Medications: These details are included in Table 2 of the manuscript. Antibiotics: None of the 16 patients in the remission (R) and mild active (mA) groups received antibiotics during the study, except for Patient No. 7. This patient developed a sudden fever during treatment, prompting hospitalization and a 2-day antibiotic course. However, infection markers were not elevated, and colonoscopy confirmed disease progression. The patient was switched from IFX to UST, and stool collection occurred more than two weeks later, minimizing antibiotic impact. Diet: All patients signed informed consent agreements to avoid yogurt, probiotics, and prebiotics during the study. While dietary factors were not explicitly recorded, we acknowledge their potential influence on gut microbiota.

In the revised manuscript, we have added a discussion of these limitations and their potential impact on the results. In the suggested research, we will implement detailed recording of these confounding factors to enhance the study’s reliability.

Changes made in the manuscript:In Page 12,Lines 338-341, it now reads: “Second, while we controlled for some confounding factors, others such as antibiotics, diet, lifestyle, and environmental influences were not rigorously monitored. These factors may have contributed to variability in the gut microbiota composition, potentially affecting the observed results.”

5、The figures are poorly designed and lack essential information. Figure 4 is particularly confusing due to its complexity. It is just a simple display of taxonomic species abundance, and it is unknown how the major proportions of microbial communities change and whether there are significant differences.

Response: Thank you for your question. Indeed, as you said, the results in Figure 4 cannot replace the analysis of differences between groups. Of course, the purpose of showing Figure 4 is not to clarify the results of the analysis of differences between groups. In fact, Figure 4 shows the composition of the bacteria with the highest abundance at each taxonomic level in different groups, in order to initially evaluate the dynamic changes of high-abundance bacteria in different groups or different disease outcome patterns (mA, R and R2A). Figure 5 is an indicator species analysis using the indicspecies package, and this method is based on the analysis of differences between groups. Therefore, we mainly use Figure 5 to explain the results of the component comparison analysis.

Changes made in the manuscript:none.

Reviewer 3 Report

Comments and Suggestions for Authors

Overall, this study provides valuable insights into microbiome dynamics in CD patients undergoing biological therapy and has the potential to contribute to precision medicine approaches in IBD management.  The introduction provides a comprehensive background on Crohn's disease (CD) and its association with gut microbiota. The discussion on biological therapies and their varying efficacy among patients is well articulated.  Introduction The authors have cited relevant studies, though they could enhance the introduction by including more recent systematic reviews or meta-analyses on gut microbiome changes in CD patients undergoing biological therapy.  

Materials and Methods The study design is appropriate for investigating longitudinal microbiome changes in CD patients receiving biological therapy. The categorization of patients into different response groups (R, mA, R2A) allows for meaningful comparative analysis. However, the relatively small sample size (n=16) may limit the generalizability of findings. The inclusion of a control group (e.g., healthy individuals or CD patients not receiving biological therapy) would have strengthened the study. The methods are described in low detail, covering patient selection, ethical considerations, sample collection, DNA extraction, but more detail are need for sequencing, and statistical analysis. Additional clarification on how confounding factors (e.g., diet, antibiotic use, disease severity at baseline) were controlled would improve methodological transparency.  

Results The results are well-structured and presented with appropriate statistical analysis. Figures and tables effectively illustrate microbial diversity, species composition, and inter-group comparisons. The inclusion of PCoA and Adonis test results strengthens the interpretation of microbiome clustering. 

Discussion The discussion could benefit from a clearer explanation of how the identified bacterial species contribute to CD progression and response to therapy. 

Conclusion The conclusions align with the study findings, emphasizing the correlation between microbiome composition and CD disease course under biological therapy. The study highlights potential biomarkers (e.g., Akkermansia muciniphila, Bifidobacterium adolescentis) for predicting therapy response. While the findings are promising, the authors should acknowledge the limitations of their study, particularly the small sample size and lack of mechanistic validation. 

Some recommendation: 

  • Expand the discussion on the clinical implications of microbial changes;  
  • Consider adding a control group for comparative analysis; 
  • Discuss potential confounding variables affecting microbiome composition; 
  • Acknowledge study limitations more explicitly in the discussion. 

Author Response

Comment :

1、Expand the discussion on the clinical implications of microbial changes

Response: In the revised manuscript, we have added a detailed discussion on the potential clinical significance of the observed microbial changes, including: The potential role of these microbial changes as biomarkers for predicting therapeutic outcomes in Crohn’s disease patients;The implications for future interventions, such as microbiota modulation through probiotics, prebiotics, or fecal microbiota transplantation, to enhance treatment efficacy. We believe these additions have strengthened the clinical relevance of our findings and provided a clearer direction for future research.

Changes made in the manuscript:In Page 13,Lines 362-370, it now reads: “The findings of this study have important clinical implications. By identifying microbial signatures associated with different treatment responses, this research could pave the way for predictive biomarkers to guide personalized therapy in IBD. For example, gut microbiota profiling could be used to stratify patients into subgroups based on their likelihood of responding to specific biologic therapies, enabling earlier and more targeted interventions. Furthermore, interventions such as fecal microbiota transplantation (FMT) or targeted modulation of key bacterial taxa could be explored as adjunct therapies to improve treatment outcomes and patient prognosis. These advances would represent a significant step toward precision medicine in IBD management.”

2、Consider adding a control group for comparative analysis

Response: The current study, our primary focus was to compare the gut microbiota of Crohn’s disease (CD) patients with different response types to biologic therapy over a longitudinal period. Therefore, we did not include a healthy control group. Additionally, due to the challenges of recruiting and retaining participants for a longitudinal study spanning over one year, especially during the COVID-19 pandemic, we were unable to incorporate a control group at this stage.

However, we fully agree with your suggestion and recognize the value of a control group in enhancing the interpretability of our findings. We draw a suggestive randomized controlled trials workflow of gut microbiome for IBD treated with biologics.

Changes made in the manuscript:In Page 12,Lines 342-345, it now reads: “Third, the absence of a healthy control group limits our ability to distinguish dis   ease-specific microbial changes from other influences. Without a control group, it is challenging to determine whether the observed microbial shifts are unique to IBD patients or reflect broader variations in gut microbiota.”

In Page 13,Lines 352-357, it now reads: “

Based on the pilot study, we suggested a scientifically rigorous clinical research protocol to investigate the association between gut microbiota and biologic therapy in IBD patients, as outlined in (Figure 6).

3、Discuss potential confounding variables affecting microbiome composition

Response:Thank you for your valuable suggestion to discuss potential confounding variables affecting microbiome composition. We fully agree that these factors are critical to consider when interpreting the results of microbiome studies. ①Disease Duration and Concomitant Medications: These details are included in Table 2 of the manuscript. ②Antibiotics: None of the 16 patients in the remission (R) and mild active (mA) groups received antibiotics during the study, except for Patient No. 7. This patient developed a sudden fever during treatment, prompting hospitalization and a 2-day antibiotic course. However, infection markers were not elevated, and colonoscopy confirmed disease progression. The patient was switched from IFX to UST, and stool collection occurred more than two weeks later, minimizing antibiotic impact. ③Diet: All patients signed informed consent agreements to avoid yogurt, probiotics, and prebiotics during the study. While dietary factors were not explicitly recorded, we acknowledge their potential influence on gut microbiota.

In the revised manuscript, we have added a discussion of these limitations and their potential impact on the results. We will also emphasize the need for future studies to carefully monitor and account for antibiotic use in similar longitudinal microbiome analyses.

Changes made in the manuscript:In Page 12,Lines 338-341, it now reads: “Second, while we controlled for some confounding factors, others such as antibiotics, diet, lifestyle, and environmental influences were not rigorously monitored. These factors may have contributed to variability in the gut microbiota composition, potentially affecting the observed results.”

4、Acknowledge study limitations more explicitly in the discussion.

Response: In the revised manuscript, we have added a dedicated section in the Discussion to explicitly address the limitations of our study, including:

Small Sample Size: The limited number of participants, particularly in the remission to active (R2A) and mild active (mA) groups (n=4 each), may reduce the statistical power and generalizability of our findings.

Lack of a Control Group: The absence of a healthy control group limits our ability to distinguish disease-specific microbial changes from other influences.

Potential Confounding Factors: Although we controlled for some variables (e.g., antibiotic use, concomitant medications), other factors such as diet and lifestyle may have influenced the results.

Lack of underlying mechanisms: we acknowledge that the current study is primarily observational and does not provide mechanistic insights into the observed microbial changes. While we have identified differences in gut microbiota composition among patients with different response types to biologic therapy, the key driver bacteria and their causal relationships with treatment outcomes remain unclear. To address this, we are planning future experiments in mouse models to explore the underlying mechanisms. Specifically, we will focus on Akkermansia muciniphila (AKK) and its associated metabolic pathways, as preliminary data suggest its potential role in modulating treatment responses. These mechanistic studies will help us move beyond descriptive observations to a deeper understanding of how specific microbial changes influence disease progression and treatment outcomes.

Changes made in the manuscript:In Page 12,Lines 330-350, it now reads:“This study has several limitations that should be acknowledged.

First, the sample size is relatively small. This was primarily due to the exploratory nature of the study, which was designed as a pilot study to preliminarily investigate the relationship between gut microbiota and treatment response in IBD patients. The small cohort size was further influenced by the rarity of Crohn’s disease, the challenges of longitudinal follow-up over one year, and the impact of the COVID-19 pandemic on patient recruitment and retention. As a result, the statistical power of the analyses may be limited, and the generalizability of the findings should be interpreted with caution.

Second, while we controlled for some confounding factors, others, such as antibiotics, diet, lifestyle, and environmental influences, were not rigorously monitored. These factors may have contributed to variability in the gut microbiota composition, potentially affecting the observed results.

Third, the absence of a healthy control group limits our ability to distinguish disease-specific microbial changes from other influences. Without a control group, it is challenging to determine whether the observed microbial shifts are unique to IBD patients or reflect broader variations in gut microbiota.

Finally, this study is primarily descriptive and does not provide mechanistic insights into the observed microbial changes, such as exploring the metabolic and immune pathways associated with key bacterial taxa (e.g., Akkermansia muciniphila), are needed to establish causal relationships and deepen our understanding of the gut microbiota’s role in treatment response.”

Round 2

Reviewer 1 Report

Comments and Suggestions for Authors

Dear Authors,

The main question to you, and what you should think about, is what you really present in this manuscript. Changes or differences? In my opinion, you mention differences of microbiome in the outcome samples (the last sample after the year of patients' treatment? It is not clearly explained in the paper as well as in the answer to my previous question). So, the study plan (interesting, and as stated, will be used in other papers) is not prepared for the present paper.

Author Response

The main question to you, and what you should think about, is what you really present in this manuscript. Changes or differences? In my opinion, you mention differences of microbiome in the outcome samples (the last sample after the year of patients' treatment? It is not clearly explained in the paper as well as in the answer to my previous question). So, the study plan (interesting, and as stated, will be used in other papers) is not prepared for the present paper.

Response: Thank you very much for your valuable feedback and for raising the important question regarding whether our study focuses on “changes” or “differences” in the gut microbiota. We appreciate your careful review and would like to clarify and address your concerns as follows:

1、Clinical Motivation and Study Objectives

The primary objective of this study stems from a critical clinical observation: while biologic agents are highly effective as first-line therapy for induction phase in Crohn’s disease (CD) patients, their long-term efficacy during the maintenance phase remains variable. CD is a lifelong condition, and over the course of years or even decades of maintenance therapy, many patients experience secondary loss of response (LOR), necessitating a switch to another biologic agent or facing severe complications such as intestinal fistulas. Currently, there are no reliable methods to predict which patients will maintain long-term remission, which will experience mild disease activity (e.g., increased stool frequency) without significant impact on quality of life, and which will develop LOR with severe disease progression.

Previous studies have explored potential predictors, including serum inflammatory markers, fecal calprotectin, and gut microbiota. However, most of these studies focused on primary non-response and were conducted during the induction phase, with limited attention to the later stages of treatment. To address this gap, we designed a pilot study to investigate whether there are distinct gut microbiota among CD patients with different response types during the maintenance phase of biologic therapy.

Our ultimate goal is to identify microbial signatures that could serve as predictive biomarkers for treatment outcomes or even as potential therapeutic targets to enhance the efficacy of biologic agents. This approach is inspired by advancements in cancer immunotherapy, where modulating the gut microbiota has been shown to convert “cold” tumors (non-responsive to immunotherapy) into “hot” tumors (responsive to immunotherapy).

We acknowledge that the initial title and abstract of our manuscript did not clearly convey this rationale. In the revised version, we have emphasized the clinical motivation behind this study.

2、Clarification of “Changes” vs. “Differences”:
Our study primarily focuses on the differences in gut microbiota composition among Crohn’s disease (CD) patients with different response types to biologic therapy. We compared the microbial profiles of patients in remission (R), mild active (mA), and moderate-severe active (R2A) groups based on their 12 samples from 1 year of collection. We acknowledge that the term “changes” may have caused confusion, and we have revised the manuscript to consistently use “differences” where appropriate.

3、Microbiome Data

We would like to clarify that the microbiome analysis was not based on one sample per patient. Instead, we collected 12 stool samples from each patient over the course of one year (monthly sampling). Therefore, the total number of samples used for the analysis is as follows:

Remission (R) group: 8 patients × 12 samples = 96 samples.

Remission to Active (R2A) and Mild Active (mA) groups: 4 patients × 12 samples = 48 samples each.

We apologize for not clearly stating this in the original manuscript. In the revised version, we have added a detailed description of the sample size and data analysis process to ensure transparency. Our raw data files is attached

4、Study Design:
We agree that the study design should align with the research question. Our study was designed as a pilot investigation to explore the differences in gut microbiota among CD patients with different treatment responses, rather than focusing on the fluctuations within individual patients over time. While we did analyze individual patient microbiota fluctuations (as shown in the figure below), we found that these fluctuations did not exhibit significant differences between groups. Therefore, our data and conclusions are tightly focused on addressing the clinical question of whether distinct microbial profiles exist among different response groups.

(The figure illustrates the fluctuations in microbial abundance for each individual patient over time.)

We acknowledge that these points were not clearly articulated in the original manuscript. In the revised version, we have added a detailed explanation to ensure clarity and transparency.

Changes made in the manuscript:

In page 1, lines 2-4 , it now reads: Title:

Gut Microbiota in Different Treatment Response Types of Crohn’s Disease Patients Treated with Biologics in Long Disease Course

In Page 1 ,lines 13-18 , it now reads: Background and Aims: Crohn’s disease (CD) is a chronic inflammatory bowel disease (IBD) with a globally increasing prevalence, partially driven by alterations in gut microbiota. Although biological therapy is the first-line treatment for CD, a significant proportion of patients experience primary non-response or secondary loss of response over time. This study aimed to explore the differences in gut microbiota among CD patients with divergent long-term responses to biologic therapy, focusing on long disease course.

In page 1 , lines 28-30, it now reads :Conclusions: Gut microbial diversity and specific bacterial significantly differed among response groups, reflecting distinct characteristics between responders and non-responders.

In page 1 , lines 31, it now reads : Keywords: Crohn’s disease; biological agents; gut microbiota; divergent treatment responses; long disease course

In page 2 , lines 72-78, it now reads : In light of these findings, we designed a pilot study to investigate whether there are distinct gut microbiota profiles among CD patients with different response types during the maintenance phase of biologic therapy.

Our study is designed to explore the differences in gut microbiota among CD patients with divergent long-term responses to biologic therapy, focusing on long disease course. Understanding distinct gut microbiota may provide crucial insights into tailored therapeutic strategies and improved management of Crohn's disease.

In page 5 , lines 130-132, it now reads : The t-test and nonparametric Mann-Whitney U test were used for comparing two groups(The dataset encompasses 12 longitudinal time points per patient).

In page 10 , lines 226-232, it now reads : In this study, we investigated the differences in fecal bacterial compositions in a prospective cohort of patients among CD treated with biological agents with different treatment responses. While prior research has explored gut microbiota in IBD patients under biological therapy, the majority of these studies focused on the initial treatment responses rather than the long-term effects of sustained therapy. This study represents valuable and comprehensive analysis of distinct gut microbiota in CD patients with divergent responding to biological treatment over long time.

In page 13, lines 362-372 , it now resds: Most studies concentrating on microbiome during the induction phase of therapy to identify precise biomarkers. Our study offers novel insights into the maintenance phase, investigating the relationship between gut microbes and disease response. We found that microbial diversity and specific bacterial populations significantly differed among response groups, reflecting distinct characteristics between responders and non-responders. Notably, we identified differences in the abundance of several bacterial species, including Akkermansia muciniphila, Faecalibacterium prausnitzii, Bifidobacterium adolescentis, Prevotella, Veillonella parvula, and Klebsiella pneumoniae, which may serve as potential biomarkers for predicting the efficacy of biologic agents in IBD. This research represents a promising step toward precision medicine.

Reviewer 2 Report

Comments and Suggestions for Authors

Can be accepted.

Author Response

Dear reviewer, 

     We would like to extend our heartfelt thanks for your invaluable feedback on our manuscript. Your thoughtful and constructive comments have played a pivotal role in enhancing the quality of our work, and we are deeply grateful for the time and expertise you dedicated to reviewing our submission.

Reviewer 3 Report

Comments and Suggestions for Authors

The Authors provided satisfying answers, even if the question of the confounding factors still remains. However, the presentation as a pilot study overcomes this and other limits.
Even if preliminary the data could have been more consistent, however, they may represent a basis for supporting other studies and researchers in the field. 
The revised manuscript is now well-enough managed and defined for the necessary changes.
The low sample size is well-defined and explained in the revised manuscript
Data regarding dietary practices is now present with an addition for using prebiotics and probiotics. It seems fine now.
The Authors added a new figure, no. 6, to the manuscript to interpret the treatment workflow, which is a nice explanation and strategy
The medication history with the proper explanation is now present in the revision.
The absence of a control group is now briefly described, which seems fine.
Table 1 still seems not well formatted; some typos e.g. in first column "tudies", please verify.

Comments on the Quality of English Language

Please verify the final English and fulfillment of the publication style

Author Response

Comment :The Authors provided satisfying answers, even if the question of the confounding factors still remains. However, the presentation as a pilot study overcomes this and other limits.
Even if preliminary the data could have been more consistent, however, they may represent a basis for supporting other studies and researchers in the field. 
The revised manuscript is now well-enough managed and defined for the necessary changes.
The low sample size is well-defined and explained in the revised manuscript
Data regarding dietary practices is now present with an addition for using prebiotics and probiotics. It seems fine now.
The Authors added a new figure, no. 6, to the manuscript to interpret the treatment workflow, which is a nice explanation and strategy
The medication history with the proper explanation is now present in the revision.
The absence of a control group is now briefly described, which seems fine.
Table 1 still seems not well formatted; some typos e.g. in first column "tudies", please verify.

Response: Thank you very much for your valuable and insightful feedback on our manuscript. We deeply appreciate your careful review and constructive suggestions, which have significantly enhanced the credibility and quality of our work.

We are pleased to hear that our responses and revisions have addressed most of your concerns. Regarding the remaining points:

Table 1 Formatting and Typos: We have thoroughly reviewed and revised Table 1, as well as the entire manuscript, to correct any formatting issues and typographical errors (e.g., “tudies” has been corrected to “studies”). These changes can be seen in the revised manuscript.

Once again, we sincerely thank you for your time and effort in reviewing our manuscript. Your precise and thoughtful comments have been invaluable in improving our work. We hope that the revised version meets your expectations and would be honored if you find it suitable for publication.

 Changes made in the manuscript:Table 1, it now reads:

studies

study type

year

Country

 No. of IBD

 patients

Medical

therapy

sample

Fecal collection Intervals

Time of follow up

Sequencing

method

result

Total

Subgroups

Rebecka
et al8

Prospective

2021

Finland

72

25CD

47UC

IFX

faecal

0 (baseline), 2, 6, and 12 weeks

1 year

16s rRNA

differed before the start of the IFX       

non-response: ClostridiaCandida

Ananthakrishnan et al9

Prospective

2017

USA

85

43UC

42CD

VDZ

faecal

0 (baseline), 6, 14, 30, and 54 weeks

1 year

Meta-

genomics

responsive CD patients: Roseburia inulinivorans and Burkholederiales  and brached chain amino-acid synthesis enriched

Aden et al10

Prospective

2019

Germany

23

13UC

10CD

IFX/ADA

faecal

0 (baseline), 2, 6, and 30 weeks

30 weeks

16s rRNA

diversity indices did not vary in remission, but in remission: butyrate and substrate were associated

Ding et al11

Prospective

2020

UK

76

76CD

IFX/ADA

faecal

0 (baseline),
3 monthly

1 year

16s rRNA

responsive CD: higher deoxycholic acid; Non-responsive CD: sulfate and glycine-conjugated primary bile acids

Effenberger

et al 12

Prospective

2021

Austria

65

43CD

22UC

IFX/ADA

faecal

0 (baseline), 12, 30 weeks

30 weeks

16s rRNA

remission: ProteobacteriaBacteroideteshigher butyrate

Colman et al13

Prospective

2023

USA

74

52 CD

21 CD

1 IBD-U

VDZ

faecal

0 (baseline),2,14 weeks, 6

month, 1year

1 year

Meta-

genomics

early response: Firmicute A. hadrus↑ abundance of butyrate

Jiang et al14

Prospective

2024

China

21

21 UC

VDZ

faecal

0 (baseline), 14 weeks

14 weeks

16s rRNA

after treatment:bifidobacterium bacteroides sartorii early remission: combined of acetamide, taurine and putrescine

Busquets et al15

Prospective

2021

Spain

38

14CD 24UC

IFX/ADA

faecal

0 (baseline) and monthly

1year

16s rRNA

greater discriminatory:M. smithi(MSM), A. muciniphila (AKK), and F. prausnitzii phylogroup2 (PHGII

Round 3

Reviewer 1 Report

Comments and Suggestions for Authors

Thank you. For me prefect. Please only prepare the references according to authors guidelines, as stated in previous review form. According to authors guidelines:

Journal Articles:
1. Author 1, A.B.; Author 2, C.D. Title of the article. Abbreviated Journal Name Year, Volume, page range.

More authors than one.  Forename as abberviation, and so one.

And imprimatur :-)

Author Response

Comment: Thank you. For me prefect. Please only prepare the references according to authors guidelines, as stated in previous review form. According to authors guidelines:

Journal Articles:
1. Author 1, A.B.; Author 2, C.D. Title of the article. Abbreviated Journal Name Year, Volume, page range.

More authors than one.  Forename as abberviation, and so one.

And imprimatur :-)

Response: Thank you very much for your thoughtful and constructive feedback on our manuscript. We deeply appreciate the time and effort you have dedicated to reviewing our work and providing insightful suggestions. Your comments have been invaluable in improving the quality and clarity of our manuscript, and we are truly grateful for your efforts.

We have carefully addressed all your comments and made the necessary revisions, including updating the reference format as per your request. We believe that these changes have significantly strengthened the manuscript and enhanced its scientific rigor.

Changes made in the manuscript:

  1. Fiocchi, C. Inflammatory Bowel Disease Pathogenesis: Where Are We?  J Gastroenterol Hepatol2015, 30, 12-8.
  2. Zuo, T.; Ng, S.C. The Gut Microbiota in the Pathogenesis and Therapeutics of Inflammatory Bowel Disease. Front Microbiol

2018, 9, 25.

  1. Nishida, A.; Inoue, R.; Inatomi, O.; Bamba, S.; Naito, Y.; Andoh, A. Gut Microbiota in the Pathogenesis of Inflammatory Bowel Disease. Clin J Gastroenterol 2018,11, 1–10.
  2. Manichanh, C.; Rigottier-Gois, L.; Bonnaud, E.; Gloux. K.; Pelletier, E.; Frangeul. L.; Nalin. R.; Jarrin, C.; Chardon, P.; Marteau. P.; Roca, J.; Dore, J. Reduced Diversity of Faecal Microbiota in Crohn’s Disease Revealed by a Metagenomic Approach. Gut2006, 55, 205–211.
  3. Wang, W.; Chen, L.P.; Zhou, R.; Wang, X.B.; Song, L.; Huang, S.; Wang, G.; Xia, B. Increased Proportions of Bifidobacterium and the Lactobacillus Group and Loss of Butyrate-Producing Bacteria in Inflammatory Bowel Disease. J Clin Microbiol2014, 52, 398–406.
  4. Andoh, A.; Imaeda, H.; Aomatsu, T.; Inatomi, O.; Bamba, S.; Sasaki, M.; Saito. Y.; Tsujikawa. T.; Fujiyama, Y.; Comparison of the Fecal Microbiota Profiles between Ulcerative Colitis and Crohn’s Disease Using Terminal Restriction Fragment Length Polymorphism Analysis. J Gastroenterol 2011, 46, 479–486.
  5. Andrews, C. N.; Griffiths, T.A.; Kaufman, J.; Vergnolle, N.; Surette, M.G.; Rioux, K.P.; Mesalazine (5-Aminosalicylic Acid) Alters Faecal Bacterial Profiles, but Not Mucosal Proteolytic Activity in Diarrhoea-Predominant Irritable Bowel Syndrome. Aliment Pharmacol Ther2011, 34, 374–383.
  6. Ventin-Holmberg, R.; Eberl.A.; Saqib, S.; Korpela, K.; Virtanen, S.; Sipponen, T.; Salonen, A.; Saavalainen, P.; Nissila, E. Bacterial and Fungal Profiles as Markers of Infliximab Drug Response in Inflammatory Bowel Disease. J Crohns Colitis2021, 15, 1019-1031.
  7. Ananthakrishnan, A.; Luo, C.W.; Yajnik, V.; Khalili, H.; Garber, J.J.; Stevens, B.W.; Cleland, T.; Xavier, R.J. Gut Microbiome Function Predicts Response to Anti-Integrin Biologic Therapy in Inflammatory Bowel Diseases. Cell Host Microbe2017,21, 603-610.
  8. Aden, K.; Rehman, A.; Waschina, S.; Pan, W.H.; Walker, A.; Lucio, M.; Nunez, A.M.; Bharti, R.; Zimmerman.J.; Bethge. J.; Schulte, B.; Schulte, D.; Franke, A.; Nikolaus, S.; Schroeder, J.O.; Vandeputte, D.; Raes, J.; Syzmczak, S.; Waetzig, G.H.; Zeuner, R.; Schmitt-kopplin, P.; Kaleta, C.; Schreiber, S.; Rosenstiel, P. Metabolic Functions of Gut Microbes Associate with Efficacy of Tumor Necrosis Factor Antagonists in Patients with Inflammatory Bowel Diseases. Gastroenterology2019, 157, 1279-1292.
  9. Ding, N S.; McDonald, J.A.K.; Perdones-Montero, A.; Rees, D.N.; Adegbola, S.O.; Misra, R.; Hendy, P.; Penez, L.; Marchesi, J.R.; Holmes, E.; Sarafian, M.H. Metabonomics and the Gut Microbiome Associated with Primary Response to Anti-TNF Therapy in Crohn’s Disease.  J Crohns Colitis 2020, 14, 1090–1102
  10. Effenberger, M.; Reider, S.; Waschina, S.; Bronowski, C.; Enrich, B.; Adolph, T.E.; Koch, R.; Moschen, A.R.; Rosenstiel, P.; Aden, K.; Tilg, H. Microbial Butyrate Synthesis Indicates Therapeutic Efficacy of Azathioprine in IBD Patients. J Crohns Colitis 2021, 15, 88–98.   
  11. Colman, R. J.; Mizuno, T.; Fukushima, K.; Haslam, D.B.; Hyams, J.S.; Boyle, B.; Noe, J.D.; Haens, G.R.D.; Limbergen, J.V.; Chun, K. Yang, J.; Denson, L.A.; Ollberding, N.J.; Vinks, A.A.; Minar, P. Real World Population Pharmacokinetic Study in Children and Young Adults with Inflammatory Bowel Disease Discovers Novel Blood and Stool Microbial Predictors of Vedolizumab Clearance. Aliment Pharmacol Ther2023, 57, 524–539.
  12. Jiang, L.L.; Liu, X.M.; Su, Y.; Chen, Y.J.; Yang, S.Z.; Ke, X.Q.; Yao, K.H.; Guo, Z.G. A Metabolomics-Driven Model for Early Remission Prediction Following Vedolizumab Treatment in Patients with Moderate-To-Severe Active Ulcerative Colitis. Int Immunopharmacol2024, 128, 111527.
  13. Busquets, D.; Oliver,L.; Amoedo, J.; Ramio-Pujol, S.; Malagon, M.; Serrano, M.; Bahi, A.; Capdevila, M.; Lluansi, A.; Torrealba, L.; Peries, L.; Chavero, R.; Gilabert, P.; Sabat, M.; Guardiola, J.; Serra-Pages, M.; Garcia-Gil, J.; Aldeguer, X.  RAID Prediction: Pilot Study of Fecal Microbial Signature with Capacity to Predict Response to Anti-TNF Treatment. Inflamm Bowel Dis2021, 27, S63-S66.
  14. Li, Y.M.; Xu, J.; Hong, Y.X.; Li, Z.J.; Xing, X.Y.; Zhufeng, Y. Z.; Lu, D.; Liu, X.; He, J.; Li, Y.H.; Sun, X.L. Metagenome-Wide Association Study of Gut Microbiome Features for Myositis. Clin Immunol2023, 255, 109738.
  15. Wingett, S.W.; Simon, S. FastQ Screen: A Tool for Multi-Genome Mapping and Quality Control. F1000Res2018, 7, 1338.
  16. Bolger, A. M.; Lohse, M.; Usadel, B. Trimmomatic: A Flexible Trimmer for Illumina Sequence Data. Bioinformatics2014 ,30, 2114–2120.
  17. Wood, D.E.; Lu, J.; Langmead, B. Improved Metagenomic Analysis with Kraken 2.Genome Biol 2019 ,20, 257.
  18. Franzosa, E. A.; Mclver, L.J.; Rahnavard, G.; Thompson. L.R.; Schirmer, M.; Weingart, G.; Lipson, K.S.; Knight, R.; Caporaso, J.G.; Segata, N.; Huttenhower, C. Species-Level Functional Profiling of Metagenomes and Metatranscriptomes. Nat Methods2018, 15, 962–968.
  19. Xu, J.; Ning, C.; Yang, S.; Zhe, W.; Na, W.; Yifan, Z.; Xinhua, R.; Yulan. L. Alteration of Fungal Microbiota after 5-ASA Treatment in UC Patients. Inflamm Bowel Dis2020, 26, 380–390.
  20. Villanueva, R.A.M.; Chen, Z.J.  Ggplot2: Elegant Graphics for Data Analysis (2nd Ed.). Measurement: Interdis Res Perspec2019, 17, 160–167. 
  21. Knights, D.; Siverberg, M.S.; Weersma, R.K.; Gevers.D.; Dijkstra, G.; Huang, H.L.; Tyler, A.D.; Sommeren, S.V.; Imhann, F.; Stepak, J.M.; Huang, H.; Vangay, P.; AI-Ghalith, G.A.; Russell. C.; Sauk, J.; Knight, J.; Daly, M.J.; Huttenhower, C.; Xavier, R.J. Complex Host Genetics Influence the Microbiome in Inflammatory Bowel Disease. Genome Med 2024, 6, 107.
  22. Lloyd-Price, J.; Arze, C.; Ananthakrishnan, A.N.; Schirmer, M.; Avila-Pacheco, J.; Poon, T.W.; Andrews, E.; Ajami, N.J.; Bonham, K.S.; Brislawn, C.J.; Casero, D.; Courtney, H.; Gonzalez, A.; Graeber, T.G.; Hall, A.B.; Lake, K.; Landers, C.J.; Mallick, H.; Plichta, D.R.; Prasad, M.; Rahnavard, G.; Sauk, J.; Shungin, D.; Vázquez-Baeza, Y.; White, R.A. 3rd; IBDMDB Investigators; Braun, J.; Denson, L.A.; Jansson, J.K.; Knight, R.; Kugathasan, S.; McGovern, D.P.B.; Petrosino, J.F.; Stappenbeck, T.S.; Winter, H.S.; Clish, C.B.; Franzosa, E.A.; Vlamakis, H.; Xavier, R.J.; Huttenhower, C. Multi-omics of the gut microbial ecosystem in inflammatory bowel diseases. Nature2019,569, 655–662.
  23. Ahmed, I.; Roy, B.C.; Khan, S.A.; Septer, S.; Umar, S. Microbiome, Metabolome and Inflammatory Bowel Disease. Microorganisms, 2016, 4,
  24. Zhuang, X.; Li, T.; Li, M.; Huang, S.; Qiu, Y.; Feng, R.; Zhang, S.; Chen, M.; Xiong, L.; Zeng, Z. Systematic Review and Meta-Analysis: Short-Chain Fatty Acid Characterization in Patients with Inflammatory Bowel Disease. Inflamm Bowel Dis2019. 25, 1751–1763. 
  25. Lewis, J.D.; Chen, E.Z.; Baldassano, R.N.; Otley, A.R.; Griffiths, A.M.; Lee, D.; Bittinger, K.; Bailey, A.; Friedman, E.S.; Hoffmann, C.; Albenberg, L.; Sinha, R.; Compher, C.; Gilroy, E.; Nessel, L.; Grant, A.; Chehoud, C.; Li, H.; Wu, G.D.; Bushman, F.D. Inflammation, Antibiotics, and Diet as Environmental Stressors of the Gut Microbiome in Pediatric Crohn’s Disease. Cell Host Microbe2017,22, 247.
  26. Kolho, K.-L.; Korpela, K.; Jaakkola, T.; Pichai, M.V.A.; Zoetendal, E.G.; Salonen, A.; de Vos, W.M. Fecal microbiota in pediatric inflammatory bowel disease and its relation to inflammation. J. Gastroenterol2015,110, 921–930.  
  27. Zhuang, X.; Tian, Z.; Feng, R.; Li, M.; Li, T.; Zhou, G.; Qiu, Y.; Chen, B.; He, Y.; Chen, M.; Zeng, Z.; Zhang, S. Fecal Microbiota Alterations Associated with Clinical and Endoscopic Response to Infliximab Therapy in Crohn’s Disease. Inflamm Bowel Dis 2020, 26, 1636–1647.
  28. Zhou, Y.; Xu, Z.Z.; He, Y.; Yang, Y.; Liu, L.; Lin, Q.; Nie, Y.; Li, M.; Zhi, F.; Liu, S.; Amir, A.; González, A.; Tripathi, A.; Chen, M.; Wu, G.D.; Knight, R.; Zhou, H.; Chen, Y. Gut Microbiota Offers Universal Biomarkers across Ethnicity in Inflammatory Bowel Disease Diagnosis and Infliximab Response Prediction.mSystems 2018, 3, e00188-17.
  29. Seong, G.; Kim, N.; Joung, J.-G.; Kim, E.R.; Chang, D.K.; Chun, J.; Hong, S.N.; Kim, Y.-H. Changes in the Intestinal Microbiota of Patients with Inflammatory Bowel Disease with Clinical Remission during an 8-Week Infliximab Infusion Cycle. Microorganisms2020,8,
  30. Sanchis-Artero, L.; Martínez-Blanch, J.F.; Manresa-Vera, S.; Cortés-Castell, E.; Valls-Gandia, M.; Iborra, M.; Paredes-Arquiola, J.M.; Boscá-Watts, M.; Huguet, J.M.; Gil-Borrás, R.; Rodríguez-Morales, J.; Cortés-Rizo, X. Evaluation of Changes in Intestinal Microbiota in Crohn’s Disease Patients after Anti-TNF Alpha Treatment. Sci Rep 2021, 11,10016.
  31. Zheng, M.; Han, R.; Yuan, Y.; Xing, Y.; Zhang, W.; Sun, Z.; Liu, Y.; Li, J.; Mao, T. The Role of Akkermansia Muciniphila in Inflammatory Bowel Disease: Current Knowledge and Perspectives. Front Immunol2023, 6,1089600.
  32. Wang, L.; Tang, L.; Feng, Y.;Zhao, S.; Han, M.; Zhang, C.; Yuan, G.; Zhu, J.; Cao, S.; Wu, Q.; Li, L.; Zhang, Z. A Purified Membrane Protein from Akkermansia Muciniphila or the Pasteurised Bacterium Blunts Colitis Associated Tumourigenesis by Modulation of CD8+ T Cells in Mice. Gut 2020, 69,1988-1997.
  33. Dunn, K.A.; Moore-Connors, J.; MacIntyre, B.; Stadnyk, A.W.; Thomas, N.A.; Noble, A.; Mahdi, G.; Rashid, M.; Otley, A.R.; Bielawski, J.P.; Van Limbergen, J.Early Changes in Microbial Community Structure Are Associated with Sustained Remission after Nutritional Treatment of Pediatric Crohnʼs Disease.Inflamm Bowel Dis 2016,22, 2853–2862.
  34. Wang, Y.; Li, L.; Chen, S.; Yu, Z.; Gao, X.; Peng, X.; Ye, Q.; Li, Z.; Tan, W.; Chen, Y. Faecalibacterium Prausnitzii-Derived Extracellular Vesicles Alleviate Chronic Colitis-Related Intestinal Fibrosis by Macrophage Metabolic Reprogramming.Pharmacol Res 2024,206, 107277.
  35. Henry, C.; Bassignani, A.; Berland, M.; Langella, O.; Sokol, H.; Juste, C. Modern Metaproteomics: A Unique Tool to Characterize the Active Microbiome in Health and Diseases, and Pave the Road towards New Biomarkers—Example of Crohn’s Disease and Ulcerative Colitis Flare-Ups.Cells 2022,11, 1340.
  36. Lopez-Siles, M.; Duncan, S.H.; Garcia-Gil, L.J.; Martinez-Medina, M. Faecalibacterium Prausnitzii: From Microbiology to Diagnostics and Prognostics. ISME J2017, 11, 841–852.
  37. YYang, Y.; Zheng, X.; Wang, Y.; Tan, X.; Zou, H.; Feng, S.; Zhang, H.; Zhang, Z.; He, J.; Cui, B.; Zhang, X.; Wu, Z.; Dong, M.; Cheng, W.; Tao, S.; Wei, H. Human Fecal Microbiota Transplantation Reduces the Susceptibility to Dextran Sulfate Sodium-Induced Germ-Free Mouse Colitis. Front Immunol2022, 13, 836542.
  38. Fan, N.; Qi, Y.; Qu, S.; Chen, X.; Li, A.; Hendi, M.; Xu, C.; Wang, L.; Hou, T.; Si, J.; Chen, S. B. Adolescentis Ameliorates Chronic Colitis by Regulating Treg/Th2 Response and Gut Microbiota Remodeling. Gut Microbes2021, 13, 1-17.
  39. K Kowalska-Duplaga, K.; Gosiewski, T.; Kapusta, P.; Sroka-Oleksiak, A.; Wędrychowicz, A.; Pieczarkowski, S.; Ludwig-Słomczyńska, A.H.; Wołkow, P.P.; Fyderek, K. Differences in the Intestinal Microbiome of Healthy Children and Patients with Newly Diagnosed Crohn’s Disease. Sci Rep2019, 9, 18880.
  40. Zhang, T.; Kayani, M.U.R.; Hong, L.; Zhang, C.; Zhong, J.; Wang, Z.; Chen, L.. Dynamics of the Salivary Microbiome during Different Phases of Crohn’s Disease. Front Cell Infect Microbiol2020, 10, 544704.
  41. Fan, X.; Lu, Q.; Jia, Q.; Li, L.; Cao, C.; Wu, Z.; Liao, M. Prevotella Histicola Ameliorates DSS-Induced Colitis by Inhibiting IRE1α-JNK Pathway of ER Stress and NF-ΚB Signaling.Int Immunopharmacol 2024,135, 112285.
  42. Rojas-Tapias, D.F.#; Brown, E.M.#; Temple, E.R.; Onyekaba, M.A.; Mohamed, A.M.T.; Duncan, K.; Schirmer, M.; Walker, R.L.; Mayassi, T.; Pierce, K.A.; Ávila-Pacheco, J.; Clish, C.B.; Vlamakis, H.; Xavier, R.J. Inflammation-Associated Nitrate Facilitates Ectopic Colonization of Oral Bacterium Veillonella Parvula in the Intestine. Nature Microbiol2022, 7, 1673–1685.
  43. Federici, S.; Kredo-Russo, S.; Valdés-Mas, R.; Kviatcovsky, D.; Weinstock, E.; Matiuhin, Y.; Silberberg, Y.; Atarashi, K.; Furuichi, M.; Oka, A.; Liu, B.; Fibelman, M.; Weiner, I.N.; Khabra, E.; Cullin, N.; Ben-Yishai, N.; Inbar, D.; Ben-David, H.; Nicenboim, J.; Kowalsman, N.; Lieb, W.; Kario, E.; Cohen, T.; Geffen, Y.F.; Zelcbuch, L.; Cohen, A.; Rappo, U.; Gahali-Sass, I.; Golembo, M.; Lev, V.; Dori-Bachash, M.; Shapiro, H.; Moresi, C.; Cuevas-Sierra, A.; Mohapatra, G.; Kern, L.; Zheng, D.; Nobs, S.P.; Suez, J.; Stettner, N.; Harmelin, A.; Zak, N.; Puttagunta, S.; Bassan, M.; Honda, K.; Sokol, H.; Bang, C.; Franke, A.; Schramm, C.; Maharshak, N.; Sartor, R.B.; Sorek, R.; Elinav, E. Targeted Suppression of Human IBD-Associated Gut Microbiota Commensals by Phage Consortia for Treatment of Intestinal Inflammation. Cell2022, 185, 2879-2898.e24.
  44. Kharaghani, A.A.; Harzandi, N.; Khorsand, B.; Rajabnia, M.; Kharaghani, A.A.; Houri, H. High Prevalence of Mucosa-Associated Extended-Spectrum β-Lactamase-Producing Escherichia Coli and Klebsiella Pneumoniae among Iranain Patients with Inflammatory Bowel Disease (IBD). Ann Clin Microbiol Antimicrob2023, 22, 86.
